# Learning to Search and Searching to Learn for Generalization in Planning

Michael Aichmüller* [1]   Yannik Hesse* [1]   Hector Geffner [1]

## Abstract

Combinatorial generalization remains a central challenge in Deep Reinforcement Learning (DRL). Classical planning provides a simple yet challenging setting to study this problem through explicit relational descriptions, without requiring learning from perception. In sparse-reward domains, standard RL exploration via real-time search is ineffective, and learning-based planning methods often rely on expert demonstrations, hindsight relabeling, or random walks from the goal state. In contrast, planners rely on best-first search methods such as A$^\star$ to solve problems from scratch. We propose a self-improving WA$^\star$ learning framework in combination with a value heuristic represented by a Relational Graph Neural Network: the heuristic guides search, and the resulting search data updates the heuristic via $Q$-learning. This loop yields heuristics that can function as general policies and solve new instances even without search, where DRL otherwise fails, as we show on puzzles such as Sokoban, Push-World, The Witness, and the 2023 International Planning Competition benchmarks. Notably, we demonstrate strong zero-shot generalization: For example, heuristics trained on Blocksworld instances with fewer than 30 blocks successfully solve instances with 488 blocks without search.

## 1. Introduction

Combinatorial generalization is a key challenge in deep reinforcement learning where the learned policies or value functions are expected to generalize out-of-distribution due to a common problem structure (Kirk et al., 2023; Lake & Baroni, 2023; Mohan et al., 2024). Classical planning is an ideal setting for studying and addressing this problem

because the common problem structure is given and does not need to be learned from pixels (Russell & Norvig, 2020; Ghallab et al., 2016; Geffner & Bonet, 2013). A planning *domain* specifies a fixed relational vocabulary and action schemas, while *instances* vary in the number of objects, the initial state, and the goal. This clean domain–instance separation induces systematic out-of-distribution shifts, including changes in branching factor and required search depth, and makes classical planning a natural testbed for *generalization* across *states*, *goals*, and *problem size* (formal definitions in Section 3).

Many challenging learning tasks in the setting of classical planning involve generalization, including learning a domain from traces drawn from hidden domain instances (Arora et al., 2018; Xi et al., 2024; Gösgens et al., 2025), and closer to the aims of this work, learning general policies and heuristics (Toyer et al., 2020; Rivlin et al., 2020; Garg et al., 2020; Karia & Srivastava, 2022; Chen et al., 2024). A general policy is a policy that can be used to solve arbitrary instances of the domain without search, and a general heuristic is an estimator of the cost to the goal which can be effectively used to search for plans in any domain instance. In recent years, DRL approaches have been used to learn general policies and heuristics over given domains (Rivlin et al., 2020; Ståhlberg et al., 2023a; Ståhlberg & Geffner, 2026). Yet, real-time search as used in these algorithms, moving iteratively from one state to a successor state (Korf, 1990; Koenig, 2001), is not an effective way to search for plans. Planners use best-first algorithms like WA$^\star$ or Greedy BFS (Richter & Westphal, 2010; Geffner & Bonet, 2013; Ghallab et al., 2016), and recent approaches have shown indeed the performance gains that can be obtained by combining Bellman updates with best-first search, which is feasible when the model is known (Agostinelli et al., 2019; Orseau & Lelis, 2021).

Early algorithms that combine full Bellman updates with real-time search include Learning Real-Time A$^\star$ (LRTA$^\star$), for deterministic MDPs, and Real-Time Dynamic Programming (RTDP), for stochastic MDPs (Korf, 1990; Barto et al., 1995). RL algorithms, which can be regarded as model-free variants, retain the real-time search, which is crucial when interacting with a world or simulator, but unnecessary when the model is known (Sutton & Barto, 2018). In this case, learning can be improved by considering forms of best-first

*Equal contribution [1]Department of Machine Learning and Reasoning, RWTH Aachen University, Aachen, Germany. Correspondence to: Michael Aichmüller <michael.aichmueller@ml.rwth-aachen.de>.

*Proceedings of the 43$^{rd}$ International Conference on Machine Learning*, Seoul, South Korea. PMLR 306, 2026. Copyright 2026 by the author(s).

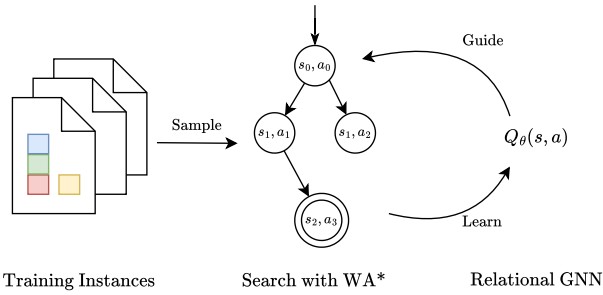

Training Instances     Search with WA*     Relational GNN

*Figure 1.* **Generalized search for planning.** During training, classical planning instances are solved with a search guided by learned $Q(s, a)$ values, which are then updated to improve performance on other instances. Unlike standard RL, (i) exploration uses best-first search (WA*) rather than stepwise real-time search, and (ii) instances vary in both initial state/goal and number of objects.

search, in particular, when the reward (the goal) is sparse. In this work, we push this idea further by learning to search for plans over the whole class of instances defined by a given planning domain. The goal is a self-improving, generalizing learning cycle: starting from a non-informative heuristic, a search for the goal on one instance yields a better heuristic, which then guides search on progressively larger instances.

In the paper, we formulate this form of self-improving generalized search (see Figure 1) that can be applied to families of path-finding problems, uniformly modeled as instances of a common planning domain. For this, general $Q$-value functions represented with relational GNNs are learned by solving training instances with search, using the states deemed relevant for updating the $Q$-functions. The self-reinforcing cycle between search and learning appears in a number of research threads in search and RL, but these forms of learning have been tied to one specific state space without generalizing systematically to others. The exceptions are recent works in classical planning for learning general policies and heuristics, yet the former rely on standard RL algorithms and hence real-time search (Rivlin et al., 2020; Ståhlberg et al., 2023a) while the latter rely on supervised learning, and do not benefit from the self-improving learning cycle of better heuristics to guide more efficient searches (Chen, 2025; Horčik et al., 2025).

There are three distinct forms of *generalization* in RL and planning: generalization to other *states*, which is required for RL to scale to large state spaces (Silver et al., 2016; 2017); generalization to other *states and goals*, as in goal-conditioned RL (Schaul et al., 2015; Andrychowicz et al., 2017); and generalization to *states, goals, and problem size*, as pursued in this work.

The paper is organized as follows: we present next the related work, background, and the generalized search for plan-

ning framework, followed by our experiments on different benchmarks and the conclusions we draw[1].

## 2. Related Work

**Planning within RL.** The Alpha family of RL algorithms (Silver et al., 2016; 2017; 2018; Schrittwieser et al., 2020) demonstrated the benefits of integrating search into the RL loop. Specifically, Monte Carlo Tree Search (MCTS), guided by value functions and policies learned via actor-critic methods, selects target states for learning, leading to more effective lookaheads and improved learning targets. While this approach performs well in challenging games, MCTS has limitations in our setting: it is ill-suited for single-goal pathfinding, and struggles in non-adversarial puzzle domains where progress depends on sparse or delayed rewards (Orseau & Lelis, 2021). AlphaZero-based general policy learning is studied in the experiments section as well, and results underline these drawbacks.

**Learning to Search.** Learning Real-Time A* (LRTA*) and Real-time Dynamic Programming (RTDP) are aimed at solving deterministic and stochastic goal-reaching MDPs with a known initial state, combining a greedy real-time search with Bellman updates (Korf, 1990; Barto et al., 1995). If the initial heuristics (cost estimators) are admissible, i.e., do not over-estimate true (expected) costs, and there are no dead-end states from which the goal is not reachable, both LRTA* and RTDP converge to an optimal policy relative to the initial state. More recently, and more closely connected to our work, different best-first search algorithms have been used inside a learning algorithm to solve hard problems like the $5 \times 5$ sliding puzzle and the Rubik's cube (Arfaee et al., 2011; Agostinelli et al., 2019; Orseau & Lelis, 2021; Hadar et al., 2026). The results are impressive but their approaches do not address size generalization.

**Learning General Policies.** General policies refer to policies that can solve any (solvable) instance of a given planning domain without search (Toyer et al., 2020; Rivlin et al., 2020; Ståhlberg et al., 2023a). Purely symbolic approaches have been developed that result in policies that can be understood and proved correct, but these methods do not scale well (Jiménez et al., 2019; Francès et al., 2021). The most recent works in this thread extend the real-time search in the DRL loop with an adaptation of hindsight experience replay (Andrychowicz et al., 2017) to the planning setting called *lifted* and *propositional HER* (Ståhlberg & Geffner, 2026), where the unachieved goals of a trace are replaced by a suitable relational description of substitute goals actually achieved. However, this form of HER is most effective

---

[1]Code and data are available in the project repository: github.com/maichmueller/generalized-search-for-planning.

when goals can be split into subgoals that can be relabeled; this assumption does not hold uniformly across planning domains and puzzles. In this paper, we build on this work, but replace the real-time search and HER with a best-first search, which is more effective in hard problems.

**Learning General Heuristics.** Since there are no perfect general policies for intractable domains like Sokoban, a different research thread focuses on learning general heuristic estimators to guide the search for plans in instances of a given domain. In these works, the heuristics $h(s)$ are learned in a supervised way from optimal values $h^*(s)$ precomputed by optimal planners (Ståhlberg et al., 2022; Horčik et al., 2025; Bai et al., 2025). Most approaches use the learned heuristic only to guide search, whereas Bai et al. (2025) additionally exploits the learned GNN representation for symmetry reduction in best-first search by pruning states through hashed GNN embeddings and actions through approximate graph automorphisms. To make learned heuristics more cost-effective, some approaches avoid GNNs altogether and instead use efficient support vector machines over relational Weisfeiler–Leman features, which are closely related to the features computable by GNNs (Chen et al., 2024). Because these methods rely on supervised learning, however, they do not benefit from the self-improving learning cycle of RL approaches.

**Search and Exploration in RL.** The real-time search used in RL algorithms is often extended with bonuses that reward novel states in the search (Burda et al., 2019; Zhang et al., 2021; Raileanu & Rocktäschel, 2020; Henaff et al., 2022). Indeed, in classical planning, a precise form of novelty, related to a formal notion of problem width, is part of the state-the-art planning algorithms as well (Geffner & Lipovetzky, 2012; Lipovetzky & Geffner, 2017; Corrêa & Seipp, 2024). Yet all state-of-the-art search algorithms in classical planning are based on best-first search, not on real-time search where current state is replaced in each step by a successor state (Korf, 1990; Koenig, 2001).

## 3. Background

**Classical Planning.** Classical planning problems are deterministic goal-reaching MDPs with extremely sparse rewards and large state spaces. They are expressed in a *language* (PDDL) that separates the description of the *domain* $\mathcal{D}$ from the concrete problem *instances* $\mathcal{I}$ of the domain. The domain specifies relation types (*predicates*) $\mathcal{P}$, where each predicate $p \in \mathcal{P}$ has arity $\mathrm{ar}(p)$, as well as *action schemas* $\mathcal{A}$ with lifted preconditions and effects defined over these relations (Geffner & Bonet, 2013; Ghallab et al., 2016; Haslum et al., 2019). Instantiating a predicate with objects yields *ground atoms* $p(o_1, \ldots, o_{\mathrm{ar}(p)})$, and instantiating an action schema with objects yields a *ground action*

$a(\bar{o})$. An instance $\mathcal{I}$ provides the object set $\mathcal{O}$, the initial state $s_0$, and a goal specification $g$. The states $s$ are represented by sets of ground atoms; namely, those which are true in the state. Preconditions and goals are conjunctions of literals, and a goal state is a state that includes all the goal atoms. A plan is a sequence of applicable actions that maps $s_0$ to a goal state.

For example, in Blocksworld, a 3-block environment is an instance with objects $\mathcal{O} = \{b_1, b_2, b_3\}$ where the states are described by means of 4 domain predicates: $on(\cdot, \cdot)$, $ontable(\cdot)$, $clear(\cdot)$, and $holding(\cdot)$. The initial state with a single tower with $b_3$ at the top, $b_2$ in the middle, and $b_1$ on the table would be $s_0 = \{on(b_3, b_2), on(b_2, b_1), clear(b_3), ontable(b_1)\}$, and the goal can be given by a single ground atom like $on(b_1, b_2)$ or by a conjunction of many such atoms. Two of the four action schemas in the domain are $stack(x, y)$, and $unstack(x, y)$, that ground in actions like $stack(b_1, b_3)$ and $unstack(b_3, b_2)$. Only the latter action is applicable in $s_0$.

**Generalized Planning and Search.** In generalized planning, we seek a general policy that can solve any domain instance. These instances vary in the initial state, goals, and number of objects, but the ground actions are instances of the same action schemas, and the states are all described in terms of the same set of predicates. This is what enables generalization while defining very precisely the scope of the generalization sought. In domains where learning with a nearly perfect compact policy is hard or impossible, such as Sokoban (Dor & Zwick, 1999), search aims at learning heuristics that speed up the search in any instance of the given domain. The approach developed in this paper serves these two purposes: it can yield nearly-perfect general policies that require no search or informed heuristic estimators.

**Search vs. Real-Time search.** In RL and in a number of algorithms like LRTA$^\star$ and RTDP, one searches for the goal using real-time search, also called agent-based search (Korf, 1990; Koenig, 2001). In this type of search, there is a current state $s$ in each iteration such that, in the next iteration, the current state $s'$ is reachable from $s$ by performing one of the applicable actions in $s$. If the *dynamic model of the problem is known*, however, a common preferred alternative is to search *best-first* as in A$^\star$, WA$^\star$, and GBFS (Russell & Norvig, 2020). Best-first search algorithms explore the space more systematically, are not affected by dead-ends, and are complete. In each iteration, they all pick the node $n$ from the search boundary that has the minimum evaluation function $f(n)$. In A$^\star$, $f(n) = g(n) + h(n)$ where $g(n)$ is the accumulated cost to reach the node $n$ from the root node, in WA$^\star$, it is $f(n) = g(n) + wh(n)$ with $w > 1$, giving thus more importance to the estimate of the cost-to-go than to the cost accumulated, while in greedy best-first search

(GBFS)–not to be confused with greedy search (real-time)–the evaluation function is $f(n) = h(n)$.

## 4. Generalized Search for Planning

In this work, we address exploration in reinforcement learning by relying on *best-first* search, as in classical planning, rather than *real-time* search, as is typical in RL. GSP (**G**eneralized **S**earch for **P**lanning) is an iterative search-and-learn scheme in which training instances are solved with weighted A* (WA*) guided by learned $Q$-values, and the resulting search data is used to improve those $Q$-values. To support generalization across states, goals, and problem sizes, we represent $Q_\theta$ with a relational graph neural network (see Section 5).

GSP maintains a parametric action-value heuristic $Q_\theta(s, a)$ and repeatedly runs WA* on sampled instances to generate experience. Each search episode expands promising state–action pairs, stores encountered transitions in a replay buffer, and (when available) attaches search-derived lower bounds on return. Q-learning updates $Q_\theta$ from this buffer, and the improved $Q_\theta$ in turn guides subsequent search episodes more effectively (cf. Figure 1).

**Search Episode.** For a sampled instance $\mathcal{E}$ with initial state $s_0(\mathcal{E})$, we run a WA* search over *state–action* nodes $(s, a)$. The algorithm maintains (i) a search tree rooted at $s_0$, (ii) a priority queue $\mathcal{F}$ (the frontier) containing candidate pairs, and (iii) a replay buffer $\mathcal{D}$ that stores tuples $(s, a, \underline{R})$ whenever a search-derived lower bound $\underline{R}$ on return is available.

We consider unit step rewards $r = -1$ without discounting, so maximizing return corresponds to finding shorter plans. Let $g(s)$ denote the accumulated return of $s$ along the current tree path from $s_0$ to $s$, its negative depth in the tree in our setting. We score each frontier pair by

$$f(s, a) = g(s) + w \, Q_\theta(s, a),$$

where $w \in \mathbb{R}$ is a weighting constant. At each expansion, we pop the pair $(s, a) \in \mathcal{F}$ with the highest score and generate the successor state $s' = a(s)$. We distinguish three types of transitions: dead-end, goal, and non-terminal. If the successor $s'$ is a non-terminal state, we insert each *previously unseen* successor pair $(s', a')$ into the tree and push it onto the frontier with score $f(s', a')$. If $s'$ is a dead-end state (no applicable actions), we assign the parent pair $(s, a)$ a fixed penalty return $R_\perp$ and store $(s, a, R_\perp)$ in $\mathcal{D}$.

If $s'$ is a goal state, we backtrack from the goal transition $(s_T, a_T)$ to the root along the discovered solution path and assign each encountered pair $(s_t, a_t)$ the actual return-to-go. This provides a lower bound $\underline{R}(s_t, a_t)$ on the optimal return from that pair, since a higher-return (shorter) solution may

exist. We store these pairs together with their lower bounds in $\mathcal{D}$. The search terminates when it reaches a goal state or exhausts its expansion budget. Pseudocode is given in Algorithm 1 in the appendix.

**Q-learning with Search-Derived Lower Bounds.** From the replay buffer, we periodically sample batches $(s, a, \underline{R}) \sim \mathcal{D}$ and regress $Q_\theta$ toward one-step Bellman targets. Let the bootstrap target be

$$\hat{y}(s, a) = -1 + \max_{a' \in \mathcal{A}(s')} Q_\theta(s', a'),$$

where $s'$ is the successor reached from $(s, a)$. We then set the learning target $y$ by pair type. For non-terminal pairs, we use $y = \hat{y}(s, a)$ (standard Q-learning), but the search provides two additional supervision signals: dead-end pairs are regressed towards the fixed penalty $y = R_\perp$, while goal-path pairs can bound the targets as

$$y = \max\{\underline{R}, \hat{y}(s, a)\}.$$

Since $\underline{R}$ comes from a concrete solution found by search, it lower-bounds the optimal return for $(s, a)$; taking the maximum prevents bootstrap targets from dropping below what search has already achieved and empirically stabilizes learning. Finally, the mean-squared error $\|Q_\theta(s, a) - y\|^2$ is minimized via stochastic gradient descent.

**Instance Selection Strategy.** Selecting which training instance to solve next is dynamic and consequential: uniform sampling wastes compute on instances that are already solved reliably or are currently out of reach. We therefore maintain three instance pools and sample from them with exponentially increasing weights: *unsolved*, *solved*, and *satisfied*. If a search successfully finds a plan, the instance is placed in *satisfied*. If additionally the expanded-nodes count equals the found plan length, we consider the instance *solved*. Intuitively, instances with suboptimal solutions provide the most informative updates, whereas instances with no solution yet or with near-best solutions contribute less beyond reinforcing existing behavior. Interestingly, if a problem is placed in *solved*, it does not mean it necessarily found the shortest path within the problem, but rather that the heuristic is so confident in guiding the search that no other nodes need to be expanded.

The GSP learning loop dynamics are visualized in Figure 2, which illustrates the number of training instances the WA* search solved when guided by learned Q-values, along with the number of nodes expanded in these searches. In the three domains (BLOCKSWORLD, SATELLITE, TRANSPORT), it is clear that as learning progresses, more training problems are solved increasingly efficiently within the given search budget. In the fourth domain shown, FLOORTILE, the

learning loop in GSP is not successful in solving all training instances and remains, on average, at high expansion numbers.

## 5. Representing $Q_\theta$ with Relational GNNs

Following earlier works on general policy learning in classical planning (Ståhlberg et al., 2022; 2023b; 2025; Aichmüller & Geffner, 2025; Chen & Thiébaux, 2024; Horčik et al., 2025), and in particular (Ståhlberg & Geffner, 2026), the Q-function is represented as a relational GNN. Planning states are indeed relational structures: a state $s$ is a set of true ground atoms[2] $p(\bar{o})$ over a finite object set $\mathcal{O}$, and the goal is a conjunction $g$ of literals. We write $\bar{o} = (o_1, \ldots, o_{\text{ar}(p)})$ for an ordered tuple of objects matching the arity of predicate $p$. In goal-conditioned RL, both the current state and the goal are required to be encoded. To this end, we form a relation set

$$\mathcal{R}_{s,g} := \{\, p(\bar{o}) \mid p(\bar{o}) \in s \cup g \,\}.$$

Since our heuristic is action-value based, we make action choices explicit during message passing by augmenting the relational input with auxiliary action-atoms. Concretely, for each applicable grounded action $a = A(\bar{o}) \in \mathcal{A}(s)$ we introduce a dedicated action object $o_a$ (one per applicable action in $s$) and an atom $A(o_a, \bar{o})$, and define

$$\mathcal{R}_A := \{\, A(o_a, \bar{o}) \mid a = A(\bar{o}) \in \mathcal{A}(s) \,\}.$$

The final input is the set $\mathcal{R} = \mathcal{R}_{s,g} \cup \mathcal{R}_A$ of relational facts over the extended object universe $\widetilde{\mathcal{O}} = \mathcal{O} \cup \{o_a : a \in \mathcal{A}(s)\}$.

We parameterize $Q_\theta(s, a)$ with a relational message-passing network that maintains a feature embedding $X_i(o) \in \mathbb{R}^d$ for each object $o$ at layer $i$, initialized to $X_0(o) = \mathbf{0}$. Given $\mathcal{R}$, embeddings are updated for $L$ layers by exchanging messages along atoms. For each atom $q = p(o_1, \ldots, o_{\text{ar}(p)}) \in \mathcal{R}$, a predicate-specific function produces *position-wise* messages

$$\left( m_{o_1}^q, \ldots, m_{o_{\text{ar}(p)}}^q \right) = \text{Comb}_p\!\left( X_i(o_1), \ldots, X_i(o_{\text{ar}(p)}) \right),$$

so that the message sent to an object depends on its argument role in the relation. An object $o$ then aggregates all incoming messages across atoms that contain it,

$$m_o = \text{Agg}\!\left( \{\, m_o^q \, : \, q \in \mathcal{R}, \, o \in q \,\} \right),$$

where Agg is permutation-invariant. Finally, the embedding is updated with a shared update function and a residual connection,

$$X_{i+1}(o) = X_i(o) + \text{Comb}_U\!\left( X_i(o), m_o \right),$$

---

[2]We follow the closed-world assumption, i.e., atoms that are not mentioned in a state $s$ are false.

for $i = 0, \ldots, L-1$.

After $L$ layers, we obtain final embeddings $\{X_L(o)\}$ that are equivariant to object permutations. We compute action-values with a single shared readout that combines the embedding of the action object with a pooled summary of the state and action objects. Let

$$\bar{X}(s, g) = \text{Pool}\!\left( \{\, X_L(o) \mid o \in \widetilde{\mathcal{O}} \,\} \right)$$

be a permutation-invariant pooling of all embeddings. For an action $a \in \mathcal{A}(s)$ with associated action object $o_a$, we define

$$Q_\theta(s, a) = \text{MLP}_Q\!\left( \left[\, X_L(o_a) \,\|\, \bar{X}(s, g) \,\right] \right),$$

where $[\cdot\|\cdot]$ denotes concatenation. Importantly, $\text{MLP}_Q$ is shared across all action schemas; action types and argument structure are expressed through the relational message passing and the resulting embedding $X_L(o_a)$. This way, the model learns a common scoring principle for grounded actions instead of separate schema-specific predictors.

We use smoothmax aggregation and parameterize the update function $\text{Comb}_U$ with a single shared multi-layer perceptron (MLP) mapping $\mathbb{R}^{2d}$ to $\mathbb{R}^d$. In contrast, we implement a separate predicate-specific MLP for each $\text{Comb}_p$, i.e., one MLP per predicate symbol $p \in \mathcal{P}$, mapping $\mathbb{R}^{d \cdot \text{ar}(p)}$ to $\mathbb{R}^{d \cdot \text{ar}(p)}$ and interpreting the output as $\text{ar}(p)$ many position-wise messages in $\mathbb{R}^d$.

## 6. Experiments

Our experiments evaluate GSP along two axes: *generalization* to unseen planning instances (varying in size, initial state, and goal) and *exploration efficiency* during training. During training, GSP generates experience via heuristic-guided weighted A$^\star$ ($w = 2$). At test time, the learned $Q_\theta$ can be used to define a greedy policy or to guide a best-first search. We therefore report results for both greedy execution (GSP$_\pi$) and WA$^\star$ ($w = 2$) guided by $Q_\theta$ (GSP$_{\text{WA}^\star}$).

We consider three types of domains: planning domains, puzzles, and the PushWorld domain, all described below. Beyond final coverage and plan-quality metrics, we analyze training dynamics by tracking the number of node expansions and instance solve rate over training time, indicating whether learning yields increasingly focused search. Figure 2 shows four exemplary domains, with the remaining domain plots found in the appendix.

**Training Setup.** The same hyperparameters were used in all experiments. In particular, we use an embedding dimension $d = 32$ and smooth-maximum aggregation. The learning rate is set to $10^{-4}$ for the R-GNN parameters and $10^{-3}$ for the readout network. Training is parallelized with

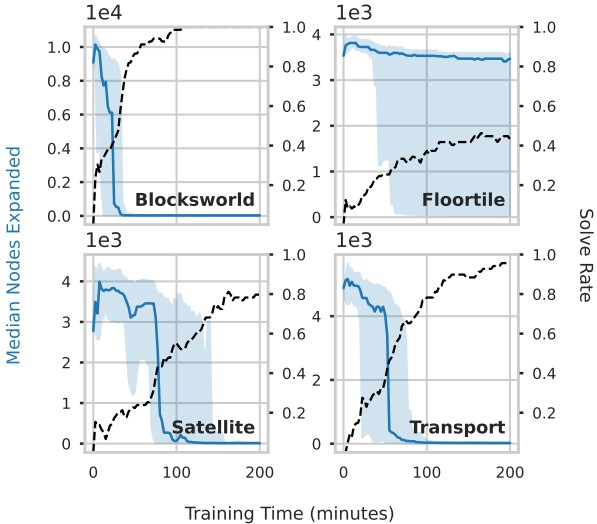

*Figure 2.* Training progress across selected IPC-learning domains. The blue line (left axis) shows the median number of expanded nodes, with shaded regions representing the 25th and 75th percentiles across all considered training instances. The black line (right axis) shows the solve rate, indicating the percentage of training instances where a goal has been found during training. Each training run lasted 12 hours (720 minutes) of which we show the first 180 minutes here.

| Domain | Objects (train/test) | Plan (train/test) |
|---|---|---|
| Blocks | 29 / 488 | 102 / 1786 |
| Transport | 34 / 453 | 57 / 1083 |
| Sokoban | $11 \times 11$ / $99 \times 99$ ($b = 3/79$) | 22 / 10546 |
| Spanner | 28 / 833 | 80 / 831 |
| Childsnack | 51 / 1326 | 33 / 879 |
| Satellite | 27 / 596 | 41 / 3428 |
| Floortile | $9 \times 3$ / $34 \times 28$ ($r = 2/26$) | 105 / 3398 |
| Miconic | 21 / 681 | 24 / 1361 |
| Ferry | 25 / 1461 | 51 / 3895 |
| Rovers | 28 / 596 | 41 / 3428 |

*Table 1.* Scaling of problem size across domains. **Objects (train/test)** reports the largest number of objects in any instance of the training and test splits (or domain-specific size parameters for grid domains where applicable: Sokoban uses grid size and boxes $b$, Floortile grid size and robots $r$). **Plan (train/test)** reports the number of steps in the (suboptimal) example solutions provided by the 2023 IPC learning benchmark. *These are not used as bounds for training.*

one learner process and five search workers that generate experience concurrently. Each search episode uses a 60 s expansion budget and worker-side batching with batch size 256. We use a FIFO replay buffer with a capacity of 40 batches. We use a target network for Q-learning and update it every 10 passes through the replay buffer (Mnih et al., 2015). A detailed overview of how we do final model selection is in the appendix.

### 6.1. Planning Domains

We use the planning benchmarks from the 2023 International Planning Competition (IPC) learning track. The benchmark comprises 10 domains, each providing 100 training instances (70 used for training and 30 for validation) and 90 test instances. The learning-track instances were designed to stress domain-independent planners by inducing large increases in object counts and applicable actions. For example, in CHILDSNACK, the hardest test instance admits 46,703,541 applicable actions in the initial state, making unguided search infeasible and placing strong pressure on heuristic quality. An overview of scaling is provided in Table 1.

**Baselines.** We compare against LIFTED HER (Ståhlberg & Geffner, 2026), a method similar to ours that exploits the relational structure using R-GNNs, while learning with real-time search, hindsight relabeling, and DQN (Mnih

et al., 2015) updates. We further compare with the domain-independent planner LAMA (Richter & Westphal, 2010), and report published solve rates for WL features (WL-f) (Chen, 2025), the state encoding of Horčík et al. (Horčík et al., 2025), and Distincter (Bai et al., 2025). We additionally evaluate an AlphaZero ($\alpha_0$) policy-value baseline using the same relational network architecture as GSP. Instead of generating data with WA*, it runs Monte Carlo Tree Search from the current state, using a network policy $\pi_\theta(\cdot \mid s)$ as a prior over applicable grounded actions and the value head to bootstrap newly expanded leaves. Root visit counts provide the policy target, and the value head is trained on the undiscounted return of the executed episode suffix. We execute the learned policy as a greedy search at test time. This baseline contrasts best-first search with the local MCTS as learning driver for generalized planning.

GSP is evaluated in two test-time modes: greedy execution induced by $Q_\theta$ (GSP$_\pi$) and WA* ($w = 2$) guided by $Q_\theta$ (GSP$_{WA*}$). A secondary experiment establishes a head-to-head comparison with LIFTED HER on the benchmarks used in their evaluation. Details on this can be found in the appendix.

**Results.** Table 2 summarizes results on IPC 2023 learning-track domains. Greedy execution performs strongly on several domains: GSP$_\pi$ reaches 100% coverage on BLOCKSWORLD, MICONIC, and SPANNER, surpassing all baselines, and achieves competitive coverage on FERRY (87%) and TRANSPORT (73%), where only LIFTED HER can outperform in both. Against WL-f and Horčík et al., GSP$_\pi$ is particularly strong on BLOCKSWORLD and MICONIC, while the alternative baselines are stronger on

| Domain | GSP$_\pi$ | | GSP$_{WA^*}$ | | Lifted HER | | LAMA | | WL-f | Horcík | Distincter | $\alpha_0$ |
| | Cov. | Steps | Cov. | Steps | Cov. | Steps | Cov. | Steps | Cov. | Cov. | Cov. | Cov. |
|---|---|---|---|---|---|---|---|---|---|---|---|---|
| blocksworld | 100% | 444 | 79% | 240 | 98% | 421 | 61% | 303 | 73% | 59% | 98% | 31% |
| childsnack | 41% | 31 | 29% | 22 | 40% | 35 | 40% | 46 | 52% | - | 71% | 0% |
| ferry | 87% | 422 | 77% | 232 | 100% | 736 | 78% | 285 | 73% | 66% | 92% | 19% |
| floortile | 20% | 54 | 28% | 57 | 0% | - | 13% | 47 | 3% | 32% | 2% | 0% |
| miconic | 100% | 490 | 98% | 268 | 90% | 562 | 100% | 301 | 98% | 68% | 100% | 28% |
| rovers | 24% | 380 | 11% | 17 | 32% | 267 | 79% | 277 | 50% | 33% | 47% | 0% |
| satellite | 61% | 134 | 33% | 18 | 56% | 394 | 100% | 163 | 57% | 40% | 53% | 0% |
| sokoban | 14% | 18 | 32% | 30 | 8% | 11 | 44% | 143 | 37% | 30% | 36% | 0% |
| spanner | 100% | 216 | 27% | 14 | 97% | 160 | 33% | 14 | 71% | 59% | 100% | 73% |
| transport | 73% | 448 | 57% | 52 | 96% | 228 | 77% | 86 | 56% | 39% | 56% | 0% |

*Table 2.* Results on 2023 IPC benchmark. We report coverage ('Cov.' fraction of 90 test instances solved) and average plan length (Steps) over solved instances for GSP as a greedy policy (GSP$_\pi$) and as a WA$^*$ ($w = 2$) heuristic (GSP$_{WA^*}$), alongside LIFTED HER and LAMA. Published coverage for WL features (WL-f) (Chen, 2025) and Horcík et al. (Horčík et al., 2025) is shown for reference. The GSP budget for each instance is limited to 10,000 expansions or one hour, whichever occurs first.

FLOORTILE and sometimes on CHILDSNACK.

Using $Q_\theta$ as a WA$^*$ heuristic yields mixed outcomes. In some domains, WA$^*$ improves coverage over greedy execution, most notably in the puzzle domains SOKOBAN ($14\% \rightarrow 32\%$) and FLOORTILE ($20\% \rightarrow 28\%$). In contrast, WA$^*$ reduces coverage on several domains where greedy execution is already strong (e.g., BLOCKSWORLD, SPANNER, and TRANSPORT), indicating that the benchmark's large branching factors can cause failures, even with effective heuristics.

The two domains, ROVERS and SATELLITE, are solved reliably only by LAMA with 79% and 100% coverage, respectively. This is an expected result due to known limitations of 1-WL expressivity on these domains, which affects all learning-based methods, but not a domain-independent planner like LAMA (Drexler et al., 2024; Horčík & Šír, 2024). CHILDSNACK is challenging for all methods, which we attribute primarily to the extreme branching factors discussed above. Distincter is the only method that performs well in this domain, suggesting that symmetry pruning partially counteracts the large branching factor.

In contrast, AlphaZero-style learning did not achieve sufficient generalization, with consistently weak results across domains except in Spanner. This lack of generalization was already visible during training, where runs often failed to generalize to all validation instances. These results support our hypothesis that MCTS is poorly suited as the main search mechanism for general policy learning in this setting. However, we do not exclude that a more targeted study on learning general policies with AlphaZero – particularly one that provides larger simulation budgets, more efficient implementations, and further algorithmic improvements from recent work on AlphaZero – may obtain stronger performance.

The difference in performance between GSP$_\pi$ and GSP$_{WA^*}$ on the IPC 2023 benchmark is attributable to two reasons. Firstly, huge branching factors in test instances degrade search, as the benchmark's design challenges search algorithms specifically. Secondly, the learned heuristic is substantially stronger as a local ranking mechanism than as a global value function on out-of-distribution data. On training problems, the learned heuristic eventually scores well enough globally that the number of expanded nodes is close to or equal to the solution length, indicating that the search is perfectly guided. However, this property deteriorates more quickly than the relative ranking of applicable actions at a state when generalizing to larger or structurally different test instances. As a result, greedy search can remain effective because it relies only on choosing the best local action, whereas WA$^*$ must order a frontier using poorly calibrated global scores. This behavior is due to encoding the state together with all applicable actions into a joint relational graph, allowing the GNN to score the actions in direct context to one another, which directly benefits action selection. However, this design does *not* encourage well-calibrated scores across states.

### 6.2. Puzzles

We next consider combinatorial puzzles that primarily test *structural* (same-size) generalization: 24-PUZZLE, SOKOBAN ($10 \times 10$, 4 boxes), and THE WITNESS ($5 \times 5$), following Orseau & Lelis (2021). We use the same training and test splits, but evaluate in a relational setting by converting the environments to PDDL. In contrast to IPC domains, instance sizes are fixed (or vary only mildly); generalization requires transferring relational reasoning to unseen configurations. Notably, unlike the IPC SOKOBAN domain, this version allows arbitrary assignment of boxes to goal locations.

| DOMAIN ⟶ | SOKOBAN / THE WITNESS / SLIDING TILE PUZZLE 5x5 | | | |
|---|---|---|---|---|
| MODEL | SOLVED | LENGTH | EXPANSIONS | TIME (S) |
| $GSP_\pi$ | 681 / 667 / 0 | 33.7 / 16.3 / – | 34 / 16 / – | 3.8 / 3.0 / – |
| $GSP_{GBFS,b=1}$ | 998 / 1000 / 0 | 38.5 / 16.2 / – | 564 / 496 / – | 59.7 / 84.0 / – |
| $GSP_{GBFS,b=32}$ | 1000 / 1000 / – | 32.6 / 14.7 / – | 1028 / 722 / – | 103 / 72.0 / – |
| $GSP_{WA^\star,w=2,b=1}$ | 1000 / 1000 / 0 | 36.0 / 16.0 / – | 207 / 548 / – | 22.1 / 94.2 / – |
| $GSP_{WA^\star,w=2,b=32}$ | 1000 / 1000 / – | 32.5 / 14.7 / – | 972 / 765 / – | 61.1 / 78.7 / – |
| LIFTED $HER_\pi$ | 309 / - / 0 | 31.2 / – / – | 31 / – / – | – / – / – |
| GBFS (†) | 914 / 290 / 0 | 37.7 / 13.3 / – | 5040 / 10128 / – | 49.2 / 44.6 / – |
| $WA^\star, w=2$ (†) | 1000 / 835 / 1000 | 35.6 / 14.2 / 130.3 | 3298 / 14305 / 1802 | 22.8 / 55.5 / 1.5 |
| $PHS^*$ (†) | 1000 / 1000 / 1000 | 37.6 / 14.4 / 222.8 | 1522 / 191 / 2764 | 11.3 / 1.7 / 3.0 |
| LevinTS (†) | 1000 / 1000 / 30 | 40.1 / 14.8 / 159.6 | 2640 / 220 / 65545 | 19.5 / 1.6 / 56.7 |
| $PHS_h$ (†) | 1000 / 1000 / 4 | 38.9 / 14.6 / 119.5 | 1962 / 222 / 58692 | 14.8 / 1.8 / 55.3 |
| DEEPCUBEA (□) | 1000 / – / – | 32.88 / – / – | 1050 / – / – | – / – / – |
| LAMA | 1000 / – / – | 51.60 / – / – | 3150 / – / – | – / – / – |

*Table 3.* Performance comparison on puzzle domains Sokoban ($10 \times 10$, 4 boxes) / The Witness ($5 \times 5$) / Sliding Tile ($5 \times 5$). 'Solved' is the number of solved instances out of 1000 per domain. Length, Expansions, and Time report averages over the solved instances in each domain ('–' if results unavailable). Sokoban models were trained on the same training problems as Orseau & Lelis (2021) and DeepCubeA (Agostinelli et al., 2019). All domains are converted to PDDL with relational encoding. LAMA results are taken from Agostinelli et al. (2019) where available. The symbol † refers to Orseau & Lelis (2021), while □ refers to Agostinelli et al. (2019). The GSP budget for each instance is limited to 100,000 expansions or five hours, whichever occurs first. We ran search modes with batch size $b = 1$ and $b = 32$, respectively, allowing a fairer comparison against baselines that reported results with $b = 32$.

**Baselines.** We compare against established puzzle solvers from Orseau & Lelis (2021), including GBFS, $WA^\star(w = 2)$, and Levin's policy-guided tree-search variants (LevinTS, $PHS^*$, $PHS_h$). We also report results for DEEPCUBEA on SOKOBAN (Agostinelli et al., 2019) and LAMA where available. For GSP, we report the modes $GSP_\pi$, $GSP_{WA^\star}$, and $GSP_{GBFS}$.

**Results.** Table 3 summarizes results across all three puzzle domains. While two methods from Orseau & Lelis (2021), $WA^\star$ and $PHS^*$, were able to solve all of the 24-PUZZLE test instances, GSP did not find a single goal path during training, and subsequently fails at test-time. Finding a solution to one of the training instances is challenging in this domain, as the training instances are not easy and initial $Q_\theta$ values are not informed. The successful baselines increase their search budgets upon failure during training and, paired with more efficient fixed-size MLPs instead of size-adaptive relational GNNs, are able to eventually find learning signals, while GSP on a fixed budget cannot. Likewise, LIFTED HER is not able to learn successfully.

On SOKOBAN, $GSP_\pi$ solves 681/1000 instances, more than double the coverage of LIFTED $HER_\pi$ (309). When used for search, $GSP_{WA^\star}$ yields perfect performance, solving all instances with 207 expansions on average, compared to 3298 expansions for $WA^\star(w = 2)$ in Orseau & Lelis (2021) and 1050 expansions for DEEPCUBEA. Similarly, $GSP_{GBFS}$ reduces the required expansions from 5040 (GBFS baseline) to 564, while simultaneously increasing coverage from 914 to 998/1000.

On THE WITNESS, both $GSP_{WA^\star}$ and $GSP_{GBFS}$ solve all instances, with expansions in the same range as LevinTS and PHS variants. This pattern is consistent with the importance of dead-end avoidance in this domain: GSP is trained not only from goal-reaching trajectories but also from explicit dead-end states. $GSP_\pi$ solves 667 instances, which shows that the learned heuristic is able to surpass even the GBFS baseline (290) in this domain, while losing only to their $WA^\star$ search (835). Results for DEEPCUBEA and LIFTED HER are not available for THE WITNESS in the cited works, and adapting their training procedures to this setting is non-trivial (reverse walks and goal relabeling depend on unavailable representation structure).

## 6.3. PushWorld

PushWorld is a Sokoban-like benchmark with sequential pushing and additional object types like composite shapes, and non-goal objects (Kansky et al., 2023). The benchmark is organized into Levels 0–5: Level 0 instances are procedurally generated, whereas Levels 1–5 are hand-designed. We use a custom PDDL formulation (released with our code). We train on Level 0 and evaluate transfer to Level 1, for which it is substantially harder to find generalizing behavior due to highly varying and larger multi-box shapes and the resulting long-horizon spatial reasoning.

**Baselines.** We compare $GSP_\pi$, $GSP_{GBFS}$, and $GSP_{WA^\star}$ against the model-free RL baselines reported by Kansky et al. (2023) (DQN, PPO) and against LAMA (Richter & Westphal, 2010).

**Results.** Table 4 summarizes results. As a greedy policy, $GSP_\pi$ solves 93/200 Level 0 test instances, substantially outperforming DQN (20/200) and PPO (11/200), while producing short plans on the solved subset. When used for search, $GSP_{WA^\star}$ solves all Level 0 instances (200/200) matching the performance of LAMA in coverage, while yielding higher quality plans on average (18 vs. 24 steps). $GSP_{GBFS}$ solves one instance fewer, but also creates worse plans on average (27 steps) than the WA* variant and LAMA.

Transfer to Level 1 remains challenging, but $Q_\theta$ still provides effective search guidance without requiring fine-tuning: $GSP_{WA^\star}$ solves 48/63 evaluable instances, with plan lengths again shorter than those of LAMA (26 vs 36 steps) on those instances solved by both methods. Yet, it requires roughly $30\times$ fewer expansions to do so. Again, $GSP_{GBFS}$ performs slightly worse, solving 4 instances fewer and with an overall worse plan quality of 47 steps.

| DOMAIN | LEVEL 0$_{ALL}$ / LEVEL 1 | | | |
|---|---|---|---|---|
| MODEL | SOLVED | LENGTH | EXPANDED | TIME (S) |
| $GSP_\pi$ | 93/5 | 13/24 | 13/24 | 2/8 |
| $GSP_{GBFS}$ | 199/44 | 27/47 | 1.2K/3.6K | 183/1314 |
| $GSP_{WA^\star}$ | 200/48 | 18/26 | 1.8K/3.8K | 273/576 |
| PPO | 11/4 | –/– | –/– | –/– |
| DQN | 20/$\approx$1 | –/– | –/– | –/– |
| LAMA | 200/60 | 24/40 | 6.9K/118K | 2/41 |

*Table 4.* PushWorld test results on Level 0$_{ALL}$ (200) and Level 1 (68). Five Level 1 instances are removed due to requiring a predicate previously unseen during training on level 0. GSP is trained on Level 0 and evaluated on both levels. 'Solved' reports solved instances; 'Length', 'Expanded', and 'Time' are averages over solved instances ('–' if unavailable), all rounded to integers. LAMA is given 90 GB memory- and 30 minute time budgets per instance. PPO and DQN results are from Kansky et al. (2023). The GSP budget for each instance is limited to 100,000 expansions or five hours, whichever occurs first.

### 6.4. Ablations

We evaluate the contribution of each training component through three ablations of GSP on the IPC planning domains: removing (i) dead-end supervision, (ii) solution lower-bound targets, and (iii) the priority-based instance sampling strategy. For each ablation, we train five seeds and evaluate the best checkpoint from each seed according to validation performance. We then report mean and standard deviation over the corresponding test results. The full results are shown in Table 7 in the appendix.

Overall, each ablated variant performs worse than the complete model on most domains, although the degradation is usually moderate. The trend is more pronounced for $GSP_\pi$ than for $GSP_{WA^\star}$. This suggests that the search-based evaluator can partially compensate for weaker learned guidance,

while the hardest domains also create floor effects that make ablation differences less visible. Among the components, dead-end supervision has the smallest impact, which is consistent with only a few IPC domains exhibiting dead-ends. Removing solution lower-bound targets causes the largest degradations, indicating that the target bounds aid in avoiding accidental regressions. The priority-based sampling strategy has a smaller effect on mean performance. More generally, the ablations show that similar results can be obtained without each component, but the complete model produces the most stable performance across domains and random seeds.

## 7. Conclusion

We introduced a simple framework for learning general $Q$-functions to guide the search for plans on arbitrary instances of a given classical planning domain. The $Q$-functions, represented by relational GNNs, are learned from instances by performing a best-first (WA*) search, informed by the $Q$-values themselves. This creates a self-improving loop in which search provides training targets and the learned heuristic improves subsequent searches. The resulting $Q$-functions generalize to other domain instances with different states, goals, and numbers of objects, and can be used at test time either as greedy policies or as heuristics for search.

The experiments cover a broad set of domains, ranging from the large, highly scaled instances of the 2023 International Planning Competition to bounded-size combinatorial puzzles and the challenging PushWorld benchmark used to evaluate RL and classical planning algorithms. The performance of GSP competes with the state of the art in almost all of these domains, while addressing this range of problems in a uniform manner, using the same architecture and hyperparameters. The results suggest that best-first search is a useful exploration mechanism for generalized reinforcement learning when the transition model is known, particularly in sparse-reward domains where real-time search struggles to discover informative trajectories.

At the same time, a limitation of the approach is that the learned $Q$-function is often stronger as a local action-ranking mechanism than as a globally calibrated heuristic over the full search frontier. Improving value generalization across states and combining best-first search with the complementary strengths of HER are natural directions for future work.

## Impact Statement

This paper presents work whose goal is to advance the field of Machine Learning. There are many potential societal consequences of our work, none of which we feel must be specifically highlighted here.

## Acknowledgements

The research has been supported by the Alexander von Humboldt Foundation with funds from the Federal Ministry for Education and Research, Germany. This project has received funding from the European Research Council (ERC) under the European Union's Horizon 2020 research and innovations programme (Grant agreement No. 885107). This project was also funded by the German Federal Ministry of Education and Research (BMBF) and the Ministry of Culture and Science of the German State of North Rhine-Westphalia (MKW) under the Excellence Strategy of the Federal Government and the Länder.

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

# A. Appendix

This appendix supplements the main paper with additional material that expands on the presented results. First, we provide a pseudocode implementation of the algorithm introduced in Section 4, shown in Algorithm 1. Second, we include further details on the experimental results and training dynamics for the IPC-Learning domains.

## A.1. Detailed Validation Strategy

Model selection is performed using a dedicated validation process. Validation is run concurrently during training, with each run saving its best checkpoint. To ensure robustness, each experiment is repeated across five random seeds, and the overall best-performing checkpoint is selected for reporting.

Within the concurrent validation loop, we periodically load the latest model parameters and evaluate greedy execution on the full validation set. Checkpoints are ranked primarily by validation coverage, with ties broken by the lowest total number of steps on solved instances, and any remaining ties resolved using the lowest RMSE on solved instances. For each experimental setting, we report the single run with the best validation score across all five seeds according to these criteria.

## A.2. Training Curves and Training Progress

The IPC-Learning domains are particularly challenging, as successful test-time performance requires strong generalization beyond the training distribution. We summarize the required scaling behavior for these domains in Table 1. Notably, our model successfully solves all BLOCKS instances, including the most difficult test case consisting of 488 blocks.

We additionally present training curves for all IPC-Learning problems in Figure 3. Most models converge on their respective training sets; however, we observe non-convergence for the FLOORTILE, ROVERS, and SPANNER domains. This lack of convergence is generally reflected in their test-time performance, with the exception of SPANNER. Although training in this domain is unstable–potentially due to forgetting or overfitting–the final model nevertheless exhibits strong generalization.

We hypothesize that this behavior is specific to the structure of the SPANNER domain. In particular, there exists a simple but suboptimal greedy strategy that involves picking up every spanner encountered. While this strategy is not optimal, it generalizes remarkably well. In contrast, attempting to learn an optimal policy eliminates this greedy behavior, which may hinder generalization. This suggests a broader challenge for domains that admit highly general-izable greedy strategies but lack an optimal strategy with similar generalization properties. Our training procedure explicitly aims to converge to the optimal general policy, which may explain the observed instability.

## A.3. Additional Direct Comparison with Lifted HER

A key baseline we consider is Lifted HER (Ståhlberg & Geffner, 2026), as it employs the same architecture and also targets generalized planning. The main distinction lies in how sparse-reward learning is handled in such domains. Unfortunately, the original work (Ståhlberg & Geffner, 2026) did not evaluate on the learning track of the IPC.

We applied our method to the domains presented in Table 6. In many training runs, the fixed validation set is unable to assess the generalization capabilities of checkpoints accurately, often selecting a sub-par model. Thus, after all fixed validation instances had been evaluated, we extended the checkpoint-selection procedure with additional benchmark test instances to obtain a more stable model selection. These instances were used only for checkpoint selection and never for gradient updates. Overall, the results are comparable to Lifted HER, with a notable advantage in DELIVERY: the original authors reported that Lifted HER struggles due to the low probability of successfully delivering two boxes, a challenge that our method addresses effectively.

## A.4. Fair comparison with Classical Heuristics

LAMA (Richter & Westphal, 2010) is a strong domain-independent classical planner. However, a direct comparison with LAMA is difficult because performance is affected not only by the heuristic itself, but also by implementation and engineering choices, especially under fixed memory and time budgets. To isolate the heuristic contribution, Table 5 compares GSP directly against the classical heuristics $h_{\mathrm{ff}}$ and $h_{\max}$ within the same execution framework. In all cases, we use WA$^\star$ with $w = 2$ and a maximum budget of $10{,}000$ node expansions.

The results show that $h_{\mathrm{ff}}$ achieves performance close to LAMA in several domains, whereas $h_{\max}$ performs substantially worse. In *miconic*, *rovers*, *satellite*, and *sokoban*, $h_{\mathrm{ff}}$ outperforms GSP. These domains appear to benefit from a more exhaustive and consistent symbolic heuristic, especially because they contain many states that are nearly identical at the symbolic level. In such cases, $h_{\mathrm{ff}}$ provides more stable heuristic estimates than GSP. Moreover, $h_{\mathrm{ff}}$ is often faster: when it fails, it typically reaches the limit of $10{,}000$ expansions, whereas GSP more often fails because of the one-hour timeout.

In the remaining domains, GSP outperforms the classical heuristics. These results suggest that GSP and $h_{\mathrm{ff}}$ capture complementary strengths. Combining the learned heuristic

with $h_{\mathrm{ff}}$, for example through a multi-queue search strategy similar in spirit to LAMA, is therefore a promising direction for future work.

---

**Algorithm 1** GSP search episode

---

**Require:** instance $\mathcal{E}$, initial state $s_0$, transition function $\mathcal{T}$,
   heuristic $Q_\theta$, weight $w$, budget $B$, dead-end bound $R_\perp$
**Ensure:** replay tuples $\mathcal{D}$ of $(s, a, \underline{R})$
1:  $\mathcal{F} \leftarrow \mathrm{Queue}(\emptyset)$
2:  $\mathcal{V} \leftarrow \{s_0\}$         *(first-discovery visitation set)*
3:  $\mathcal{D} \leftarrow \emptyset$
4:  **for all** $a \in \mathcal{A}(s_0)$ **do**
5:    $\mathrm{PARENT}(s_0, a) \leftarrow \perp$
6:    $g(s_0) \leftarrow 0$
7:    push $(s_0, a)$ into $\mathcal{F}$ with priority $g(s_0) + wQ_\theta(s_0, a)$
8:  **end for**
9:  **for** $t = 1$ to $B$ **do**
10:    **if** $\mathcal{F} = \emptyset$ **then**
11:      **break**
12:    **end if**
13:    pop $(s, a)$ with maximal priority from $\mathcal{F}$
14:    $s' \leftarrow \mathcal{T}(s, a)$
15:    **if** $s'$ is goal **then**
16:      $R \leftarrow 0$        *(suffix return along goal path)*
17:      **while** $(s, a) \neq \perp$ **do**
18:        $R \leftarrow R - 1$
19:        **if** $(s, a, -\infty) \in \mathcal{D}$ **then**
20:          remove $(s, a, -\infty)$ from $\mathcal{D}$
21:        **end if**
22:        add $(s, a, R)$ to $\mathcal{D}$
23:        $(s, a) \leftarrow \mathrm{PARENT}(s, a)$
24:      **end while**
25:      **return** $\mathcal{D}$
26:    **end if**
27:    **if** $\mathcal{A}(s') = \emptyset$ **then**
28:      add $(s, a, R_\perp)$ to $\mathcal{D}$; **continue**
29:    **end if**
30:    add $(s, a, -\infty)$ to $\mathcal{D}$ *(unbounded bootstrap sample)*
31:    **if** $s' \notin \mathcal{V}$ **then**
32:      $\mathcal{V} \leftarrow \mathcal{V} \cup \{s'\}$;
33:      **for all** $a' \in \mathcal{A}(s')$ **do**
34:        $\mathrm{PARENT}(s', a') \leftarrow (s, a)$
35:        $g(s') \leftarrow g(s) - 1$
36:        push $(s', a')$ into $\mathcal{F}$ with priority $g(s') + wQ_\theta(s', a')$
37:      **end for**
38:    **end if**
39: **end for**
40: **return** $\mathcal{D}$

---

| Domain | $\text{GSP}_\pi$ | | $\text{GSP}_{\text{WA}^\star}$ | | | $\text{WA}^\star{}_{h_{\text{ff}}}$ | | | $\text{WA}^\star{}_{h_{\text{max}}}$ | | |
|---|---|---|---|---|---|---|---|---|---|---|---|
| | Cov. | Steps | Cov. | Exp. | Steps | Cov. | Exp. | Steps | Cov. | Exp. | Steps |
| blocksworld | 100% | 444 | 79% | 29 | 29 | 12% | 2159 | 34 | 3% | 702 | 13 |
| childsnack | 41% | 31 | 29% | - | - | 0% | - | - | 0% | - | - |
| ferry | 87% | 422 | 77% | 110 | 110 | 67% | 480 | 115 | 6% | 1076 | 11 |
| floortile | 20% | 54 | 28% | 42 | 42 | 8% | 3599 | 43 | 0% | - | - |
| miconic | 100% | 490 | 98% | 566 | 268 | 100% | 840 | 288 | 20% | 1802 | 12 |
| rovers | 24% | 380 | 11% | 245 | 17 | 31% | 82 | 17 | 6% | 2053 | 11 |
| satellite | 61% | 134 | 33% | 234 | 18 | 67% | 24 | 18 | 6% | 1248 | 7 |
| sokoban | 14% | 18 | 32% | 556 | 29 | 34% | 194 | 31 | 23% | 607 | 18 |
| spanner | 100% | 216 | 27% | 981 | 14 | 33% | 64 | 12 | 32% | 483 | 12 |
| transport | 73% | 448 | 57% | 29 | 28 | 34% | 728 | 34 | 7% | 1573 | 8 |

*Table 5.* Results on 2023 IPC benchmark. We report coverage ('Cov.' fraction of 90 test instances solved) and average plan length (Steps) over solved instances for GSP as a greedy policy ($\text{GSP}_\pi$) and as a $\text{WA}^\star$ ($w = 2$) heuristic ($\text{GSP}_{\text{WA}^\star}$), alongside $h_{\text{ff}}$ and $h_{\text{max}}$. For $\text{GSP}_{\text{WA}^\star}$ and $\text{WA}^\star{}_{h_{\text{ff}}}$, expanded nodes (Exp.) and plan length (Steps) are reported only on problems solved by both $\text{GSP}_{\text{WA}^\star}$ and $\text{WA}^\star{}_{h_{\text{ff}}}$. For $\text{WA}^\star{}_{h_{\text{max}}}$, Exp. and Steps are reported on problems solved by both $\text{GSP}_{\text{WA}^\star}$ and $\text{WA}^\star{}_{h_{\text{max}}}$. The search budget (also for $h_{\text{ff}}$ and $h_{\text{max}}$) for each instance is limited to 10,000 expansions or one hour, whichever occurs first.

| Domain | $\text{GSP}_\pi$ | | $\text{GSP}_{\text{WA}^\star}$ | | Lifted HER | | Propositional HER | | LAMA | |
|---|---|---|---|---|---|---|---|---|---|---|
| | Cov. | Steps | Cov. | Steps | Cov. | Steps | Cov. | Steps | Cov. | Steps |
| blocks | 100% | 97.9 | 100% | 96.5 | 100% | 96.7 | 100% | 107.3 | 100% | 232.5 |
| childsnack | 79% | 89.3 | 34% | 59.4 | 56% | 77.8 | 100% | 92.6 | 100% | 98.4 |
| delivery | 93% | 293.3 | 42% | 134.7 | 12% | 292.2 | 11% | 167.0 | 99% | 276.1 |
| gripper | 100% | 238.0 | 74% | 202.6 | 100% | 238.0 | 100% | 239.0 | 100% | 238.0 |
| miconic | 100% | 158.0 | 100% | 158.0 | 100% | 160.0 | 100% | 158.4 | 100% | 195.6 |
| reward | 75% | 119.3 | 99% | 94.2 | 58% | 95.9 | 72% | 91.6 | 99% | 121.8 |
| spanner | 95% | 95.3 | 0% | 0.0 | 100% | 95.5 | 100% | 95.5 | 0% | 0.0 |
| visitall | 94% | 562.4 | 41% | 331.1 | 100% | 453.3 | 88% | 392.6 | 100% | 530.0 |

*Table 6.* Benchmark results on the same data as (Ståhlberg et al., 2025). We report coverage (Cov.) and average plan length (Steps) for GSP as a greedy policy ($\text{GSP}_\pi$) and as a $\text{WA}^\star$ ($w = 2$) heuristic ($\text{GSP}_{\text{WA}^\star}$), compared against LIFTED HER and PROPOSITIONAL HER (Ståhlberg et al., 2025) and LAMA.

| GSP$_\pi$ | *Complete* | | *No dead-end supervision* | | *No solution bounds* | | *No sampling strategy* | |
|---|---|---|---|---|---|---|---|---|
| Domain | Cov. | Steps | Cov. | Steps | Cov. | Steps | Cov. | Steps |
| blocksworld | 83% ± 24% | 410 | 89% ± 16% | 368 | 82% ± 24% | 343 | 63% ± 16% | 284 |
| childsnack | 33% ± 15% | 71 | 53% ± 12% | 279 | 44% ± 10% | 38 | 33% ± 16% | 36 |
| ferry | 92% ± 9% | 605 | 83% ± 11% | 435 | 72% ± 17% | 328 | 89% ± 19% | 801 |
| floortile | 19% ± 5% | 62 | 29% ± 6% | 93 | 19% ± 8% | 66 | 33% ± 6% | 134 |
| miconic | 96% ± 9% | 309 | 84% ± 15% | 195 | 74% ± 25% | 193 | 95% ± 9% | 251 |
| rovers | 26% ± 4% | 491 | 24% ± 5% | 475 | 31% ± 4% | 716 | 26% ± 2% | 692 |
| satellite | 44% ± 10% | 331 | 53% ± 6% | 241 | 47% ± 9% | 308 | 48% ± 5% | 276 |
| sokoban | 16% ± 3% | 19 | 15% ± 1% | 20 | 19% ± 2% | 20 | 17% ± 2% | 23 |
| spanner | 97% ± 4% | 190 | 72% ± 21% | 186 | 36% ± 5% | 22 | 55% ± 29% | 99 |
| transport | 85% ± 15% | 407 | 76% ± 11% | 542 | 73% ± 10% | 184 | 80% ± 7% | 424 |

| GSP$_{WA^\star}$ | *Complete* | | *No dead-end supervision* | | *No solution bounds* | | *No sampling strategy* | |
|---|---|---|---|---|---|---|---|---|
| Domain | Cov. | Steps | Cov. | Steps | Cov. | Steps | Cov. | Steps |
| blocksworld | 72% ± 4% | 196 | 78% ± 5% | 236 | 78% ± 7% | 235 | 71% ± 5% | 188 |
| childsnack | 23% ± 14% | 23 | 27% ± 15% | 24 | 29% ± 9% | 22 | 27% ± 15% | 24 |
| ferry | 67% ± 0% | 110 | 67% ± 0% | 110 | 66% ± 5% | 111 | 68% ± 2% | 119 |
| floortile | 32% ± 2% | 59 | 34% ± 2% | 66 | 26% ± 9% | 60 | 35% ± 2% | 72 |
| miconic | 81% ± 9% | 141 | 84% ± 11% | 167 | 81% ± 17% | 160 | 89% ± 11% | 199 |
| rovers | 8% ± 3% | 20 | 12% ± 4% | 19 | 11% ± 4% | 18 | 11% ± 3% | 19 |
| satellite | 27% ± 9% | 15 | 33% ± 0% | 18 | 29% ± 6% | 15 | 33% ± 1% | 17 |
| sokoban | 33% ± 1% | 30 | 32% ± 0% | 30.0 | 33% ± 1% | 29 | 33% ± 1% | 30 |
| spanner | 26% ± 2% | 14 | 30% ± 4% | 14 | 25% ± 2% | 12 | 31% ± 2% | 14 |
| transport | 56% ± 6% | 51 | 50% ± 2% | 44 | 46% ± 2% | 35 | 50% ± 3% | 41 |

*Table 7.* IPC-2023 test results for GSP$_\pi$ and GSP$_{WA^\star}$ across training variants. *Cov.* denotes the fraction of the 90 test instances solved, reported as mean ± standard deviation over independently trained models. *Steps* denotes the average plan length over solved instances, averaged across models. GSP$_{WA^\star}$ evaluates the learned model with WA$^\star$ search using $w = 2$.

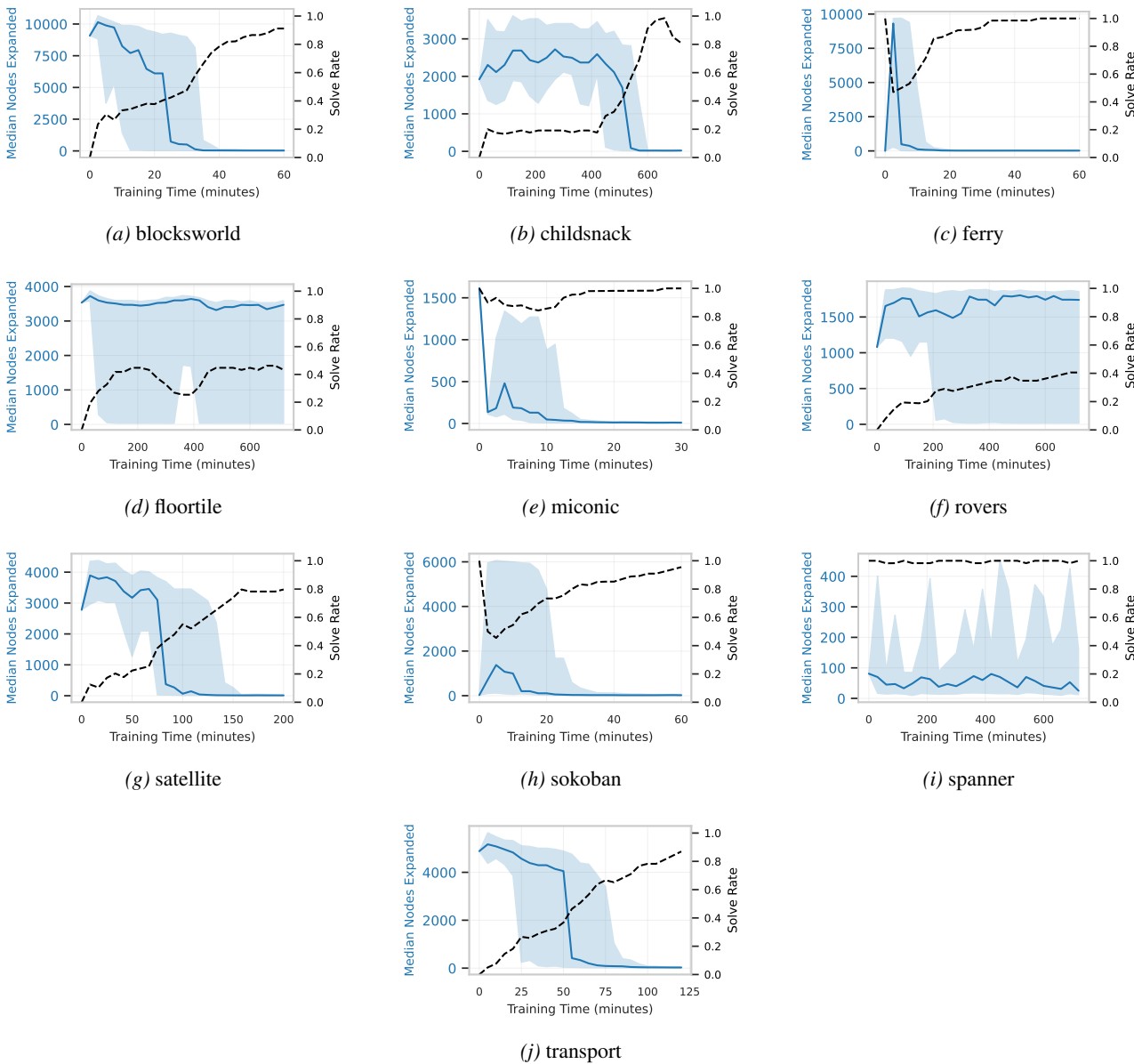

*Figure 3.* Training progress across all IPC-learning domains. The blue line (left axis) shows the median number of expanded nodes, with shaded regions representing the 25th and 75th percentiles across all instances. The black line (right axis) shows the solve rate, indicating the percentage of problems solved at each point during training. Each training run lasted 12 hours (720 minutes), and we truncated the plots for runs considered converged.

