# OpenReview forum: "Learning to Search and Searching to Learn for Generalization in Planning"
_ICML.cc/2026/Conference — ICML 2026 regular_

### Official Review · Reviewer_vean · 2026-02-22

**Soundness:** 2
**Presentation:** 3
**Significance:** 2
**Originality:** 2
**Overall Recommendation:** 4
**Confidence:** 3

**Summary:**

This paper proposes Generalized Search for Planning (GSP) to tackle combinatorial generalization in RL. The real time exploration typically used by RL is replaced by a best-first search, with heuristic values derived from Q-values represented by a relational GNN. During training, a budgeted search is performed using the current Q-value heuristic. The training signal used to update the Q-value network are solution paths for lower-bounds on the grounded value, and dead-end penalties. The evaluations performed consider using the learned Q-value network as both greedy policies and as a heuristic guiding a tree search.

**Compliance With Llm Reviewing Policy:**

Affirmed.

**Final Justification:**

The author's clarified the doubts I had with respect to fair comparisons and ablations, however I still share similar concerns as reviewer jNZG.

**Key Questions For Authors:**

1. I believe the results which were carried over by Orseau & Lelis, 2021 (i.e. GBFS, WA*, PHS*, LevinTS, $\text{PHS}\_h$) use a batched version of search with B=32, where child nodes are batched for inference (heuristic/policy) before their costs are computed and added back into the open list. This is done to increase GPU utilization (important during training) but also means that more expansions are used than if you were to use B=1 at test time. Can you clarify what is the inference batch size GSP uses, and whether you are comparing methods which use batched inference at test time against those which do not? This is important when comparing whether GPS actually solves the same amount of problems in fewer expansions or not.
2. Can you provide ablations over each of your components so its a bit more clear where gains/losses are coming from, i.e., dead-end penalty, solution lower bound, your problem selection scheme vs small to increasingly large budget used in the Bootstrap approach?
3. The appendix states 5 seeds are used but reported metric tables only show the best run. With some domains showing failures, I'd like to know the mean/variance for the reported results.
4. Can you explain the procedure for how the dead-end value $R_\bot$ is chosen? Too small or large of a value will have an impact on the Q-values learned over all state/action pairs. Do you have ablations over this?
5. For the planning domain results, there are some large differences between the performance of the greedy $\text{GSP}_{\pi}$ versus $\text{GSP}\_{\text{WA*}}$. Can you help explain where the failure mode is (e.g. an inaccurate heuristic, branching factor, etc.). I think some additional experiments here are warranted, such as performing a top-k restriction on WA*, or using a heuristic of $h(s) = - \text{max}_a Q(s,a)$, would help with the analysis.

**Limitations:**

yes

**Strengths And Weaknesses:**

**Strengths**
- The overall method is coherent and well motivated, highlighting how search can help with replacing the real-time RL exploration
- Strong baselines and domains are used for experimental validation
- Empirical results support claim that GSP can scale to out-of-distribution planning domains at test time
- The training curves support the self-improving premise

**Weaknesses**
- The outer training loop of selecting a curriculum of easy to harder problems while performing search to gather training data to learn a heuristic is very close to the Bootstrap Method [1].  A comparison isolating your contribution of problem selection against this would be helpful to highlight your contributions.
- The empirical results seem mixed when it comes to comparing the performance against the other baselines
- There are clear domain failures where GSP fails to find any solution (sliding tile puzzle)
- The compute/budget and baseline comparability is hard to interpret cleanly, especially when importing results from previous papers and using wall clock time.

**Misc**
- Orseau & Lelis, 2021 (the AAAI version cited) uses WA* with w=1.5. The arXiv version has additional experimental results for w=2. For readers who want to go to the source to copy their experimental values, you should note where to find them.

[1] Arfaee, S. J., Zilles, S., and Holte, R. C. Learning heuristic functions for large state spaces. Artificial Intelligence, 175(16-17):2075–2098, 2011.

---

> ### Author Rebuttal · Authors · 2026-03-31
>
> Thank you for your thoughtful review. We address your questions below.
>
> **Q1 (Batched inference and fair comparison):**
> We also investigated batching and found that it provides a significant performance boost in terms of states evaluated per second. However, to present clean and interpretable results, we used a batch size of 1 in our main experiments. To address your concern about a fair comparison with the results of Orseau & Lelis (2021), we reran our evaluation suite with a batch size of 32. The results are as follows:
>
> | Model | Solved | Steps | Expansions | Time (s) |
> |---|---:|---:|---:|---:|
> | *Sokoban / The Witness* |  |  |  |  |
> | GSP_GBFS, b=1 | 998 / 1000 | 38.5 / 16.2 | 564 / 496 | 59.7 / 84.0 |
> | GSP_WA*, w=2, b=1 | 1000 / 1000 | 36.0 / 16.0 | 207 / 548 | 22.1 / 94.2 |
> | GSP_GBFS, b=32 | 1000 / 1000 | 32.6 / 14.7 | 1028 / 722 | 103 / 72.0 |
> | GSP_WA*, w=2, b=32 | 1000 / 1000 | 32.5 / 14.7 | 972 / 765 | 61.1 / 78.7 |
> | GBFS (†) | 945 / 290 | 37.7 / 13.3 | 7284 / 10128 | 49.2 / 44.6 |
> | WA*, w=2 (†) | 1000 / 835 | 35.6 / 14.2 | 3298 / 14305 | 22.8 / 55.5 |
>
> Using a batch size of 32 at test time increases our node expansions, but they remain well below those of Orseau & Lelis (2021). At the same time, our mean plan length decreases notably for both Sokoban and The Witness.
>
> **Q2 (Ablations):**
> We conducted ablations removing the dead-end penalty, the priority-based problem sampling, and the solution lower bounds on three IPC planning domains: Blocksworld, Childsnack, and Spanner. Due to compute constraints we could not run ablations on all domains in time now.
> We also noticed a data error in our Childsnack results in the paper. We apologize for this and provide the correct results below as well.
>
> | Domain | Experiment | Greedy Cov. (%) | Greedy Steps | WA* (w=2) Cov. (%) | WA* (w=2) Steps | WA* (w=2) Exp. |
> |---|---|---:|---:|---:|---:|---:|
> | Blocksworld | full | 100 | 443.6 | 79 | 240.4 | 898.4 |
> | Blocksworld | ablation-deadend | 98 | 430.3 | 78 | 232.6 | 450.5 |
> | Blocksworld | ablation-nosampling | 96 | 408.3 | 76 | 211.6 | 393.8 |
> | Blocksworld | ablation-pathbounds | 81 | 299.3 | 73 | 195.6 | 457.2 |
> | Childsnack | full (corrected) | 69 | 60.0 | 38 | 26.6 | 84.1 |
> | Childsnack | ablation-deadend | 52 | 175.0 | 59 | 46.4 | 225.7 |
> | Childsnack | ablation-nosampling | 72 | 54.2 | 67 | 53.9 | 65.3 |
> | Childsnack | ablation-pathbounds | 47 | 34.1 | 47 | 34.1 | 77.0 |
> | Spanner | full | 100 | 216.4 | 27 | 14.2 | 981.5 |
> | Spanner | ablation-deadend | 80 | 184.5 | 33 | 13.9 | 15.4 |
> | Spanner | ablation-nosampling | 34 | 15.9 | 27 | 12.9 | 850.6 |
> | Spanner | ablation-pathbounds | 76 | 220.1 | 23 | 11.4 | 1419.4 |
>
> In general, the full model, combining all components, performs best on Blocksworld and Spanner. For Childsnack, we found that one of the five runs without priority sampling achieved a higher solve rate than the default; however, variance on this domain is high. We report both the mean coverage on test instances and standard deviation across seeds:
>
> | Experiment | Mean Cov. (%) | Std. Dev. |
> |---|---:|---:|
> | Greedy [nosampling] | 0.44 | 0.23 |
> | WA*2 [nosampling] | 0.26 | 0.24 |
> | Greedy [full] | 0.62 | 0.07 |
> | WA*2 [full] | 0.37 | 0.06 |
>
> The version without priority sampling exhibits considerably more variance across seeds, whereas adding priority sampling stabilizes performance. The key takeaway from the ablation is that removing the solution path bounds causes the largest drop in coverage, for example from 100% to 81.1% on Blocksworld greedy, while removing the dead-end penalty and sampling each have more moderate effects that vary by domain. We will include a more detailed report of all ablation results in the final paper.
>
> **Q3 (Dead-end penalty):**
> The dead-end penalty for all domains was set to a fixed value of -200. We verified that no training instance had an optimal solution cost exceeding this magnitude, which motivated the choice. For very simple problems with many dead ends, we found that choosing a penalty closer to the worst-case optimal path length worked better, but we decided to stick with the same hyperparameters overall. We attribute this to training with the MSE loss, which can become unstable when the dead-end penalty is too extreme relative to the Q-value range.
>
> **Q4 (Greedy vs. WA\* failure modes):**
> We did experiment with beam search and other restricted search variants, but these modifications do not address the core reason why global search underperforms on certain domains. We discuss this in more detail in our response to Reviewer 2.
>
> **Regarding novelty and the Bootstrap Method:**
> We refer to our response to Reviewer 3 for a detailed discussion.
>
> We hope these responses address your concerns.

---

> > ### Author Rebuttal · Reviewer_vean · 2026-04-03
> >
> > The author's clarified the doubts I had with respect to fair comparisons and ablations, however I still share similar concerns as reviewer jNZG. I have updated my score to a 4.

---

### Official Review · Reviewer_jNZG · 2026-02-23

**Soundness:** 4
**Presentation:** 4
**Significance:** 3
**Originality:** 1
**Overall Recommendation:** 2
**Confidence:** 4

**Summary:**

The paper proposes a curriculum style RL approach to learn value functions guiding search in classical planning. Like other works before, the paper exploits the symbolic description of the planning problem domain in order to represent the value function as a relational neural network in a problem-generic manner. This allows knowledge transfer from one to another problem of the same problem domain. The authors propose to train this value function via a form of Q-learning that is inspired by early works on heuristic dynamic programming algorithms. More specifically, the value function is trained by iterating rollouts, generating a search tree for a given training problem using the current value function snapshot for guidance, and value-function parameter improvement, where batches of the collected search trees are sampled and value function is updated using the Bellman equation and a mean-square error loss. A large empirical evaluation demonstrates the effectiveness of the approach.

**Compliance With Llm Reviewing Policy:**

Affirmed.

**Final Justification:**

I appreciate the authors' response, which clarified some misunderstandings I initially had. Nevertheless, my overall assessment remains the same: the paper does not provide any novel and significant contribution justifying a full conference paper.

The sole contribution lies in a) a training method, which learns Q values from a partially expanded search tree, and b) an instance selection strategy, which loosely follows the idea of curriculum learning. The Q function representation was taken from prior work [e.g., 1]. Methods doing a) and b) existed prior to this work.

Specifically, regarding a), the presented method is closely related to "dynamic programming" search (RTDP, AO*, etc.), one of the more fundamental differences is using an approximate function representation instead of Q table. A comparison to such baselines is not provided. Moreover, Q-function like learning from an incomplete search state has been considered in prior work before too (although not in the exact way as proposed in the paper, e.g., [2] learn functions predicting the number of expansion of search, which doesn't require a computed solution).

b) is essentially boils down to a simpler form of bootstrapping [3], where one generates and tries to learn from incrementally harder problems.

[1] Simon Ståhlberg, Blai Bonet, Hector Geffner: Learning General Optimal Policies with Graph Neural Networks: Expressive Power, Transparency, and Limits. ICAPS 2022: 629-637
[2] Patrick Ferber, Florian Geißer, Felipe W. Trevizan, Malte Helmert, Jörg Hoffmann: Neural Network Heuristic Functions for Classical Planning: Bootstrapping and Comparison to Other Methods. ICAPS 2022: 583-587
[3] Shahab Jabbari Arfaee, Sandra Zilles, Robert C. Holte: Bootstrap Learning of Heuristic Functions. SOCS 2010: 52-60

**Key Questions For Authors:**

1. Do you have any insights as to how the WA*-based learning fails, where LRTA does not, and vice versa?
2. How did you determine the w parameter? How sensitive is the approach in different values of w?

**Limitations:**

Yes

**Strengths And Weaknesses:**

The paper is written clearly and is easy to follow. The topic is generally relevant for ICML. The paper comes with an extensive empirical evaluation, over various different benchmark sets, and covers all state-of-the-art methods in this area. The empirical results show improvements over the state of the art in some benchmark domains.

The main problem is that the paper neither introduces any novel ideas, nor makes any other notable contribution over the related literature. The main contribution is rather an engineering one: the paper adopts almost 1-to-1 a recently proposed method for learning value functions, just replacing the training data generation pipeline with another well-known search algorithm. While empirical results are good, this addition would be rather suited for an extension of that mentioned prior work to a journal rather than as a standalone conference paper. For a conference, the contributions are too incremental.

Moreover, the experiments section lack truly interesting insights. Too much space focuses on reading off results that are visible in the tables and the plots anyway. For example, it would be interesting to pinpoint the particular reasons and characteristics that make the proposed approach work or not work in domains where the competitors don't work or work, respectively.

---

> ### Author Rebuttal · Authors · 2026-03-31
>
> We appreciate the careful reading and understand the concern about incrementality. We will attempt to address all concerns in the following.
>
> **Q1 (WA\* vs LRTA\*):** LRTA\* is discussed in Section 2 as historical context, but it is not a baseline in our evaluation nor in any recent work in this area. The relevant distinction is between real-time search, as used by all RL-based baselines including Lifted HER, and best-first search, as used by GSP. Real-time search commits to a single trajectory and struggles with dead-ends and sparse rewards. Best-first search maintains a global frontier and can back out of dead-ends systematically. Our contribution is not the choice of WA\* itself, but the novel training procedure built around the full search tree it produces, as discussed below.
>
> **Q2 (w parameter):** We use $w = 2$ uniformly. Performance was largely insensitive to the specific value of $w > 1$. We want $w > 1$ because this leads to convergence of $|plan length| = |expansions|$. We will include an ablation in the revision.
>
> **Clarifying the Core Contribution**
>
> The contribution is not the choice of search algorithm, but how we extract learning signal from search.
>
> Consider the following example. From state $s$, action $a$ leads to the goal, while action $b$ leads to a dead-end subtree. If the heuristic incorrectly favors $b$, search expands the subtree under $b$ before finding the goal via $a$. Prior methods, such as Arfaee et al. (2011) and Orseau & Lelis (2021), train only on solution paths, so $b$ receives no signal. The agent then repeats the same mistake in every episode.
>
> GSP updates the entire expanded frontier. Action $b$ is explicitly penalized, so the next episode skips that subtree. This is what lets the learned Q-function act as a standalone policy on new, larger instances.
>
> Three specific novelties follow:
>
> 1. *Full search-tree updates:* Every expanded state-action pair receives a learning signal, including negative signal for unproductive branches, which lets search converge to near-perfect guidance with expanded nodes approximately equal to plan length.
>
> 2. *Search-boundary supervision:* Q-targets are clamped to never fall below demonstrated returns, and dead-ends receive explicit penalties. Prior methods discard this information.
>
> 3. *Forward curriculum without invertibility:* Bootstrap and DeepCubeA require backward random walks from goal states and therefore assume invertible operators and apriori known goal states. GSP uses forward-generated instances with dynamic prioritization, thereby handling conjunctive goals and non-invertible actions naturally.
>
> **Relation to Lifted HER**
>
> Lifted HER (Ståhlberg & Geffner, 2026) is our closest baseline and shares the same R-GNN architecture. Much of that paper is dedicated to addressing the same sparse-reward problem that we tackle: real-time search alone cannot reliably reach goals, so they introduce elaborate hindsight relabeling to generate training signal from failed episodes. GSP offers a simpler solution to the same problem: best-first search reaches goals directly, removing the need for relabeling and its assumption that goals decompose into independent subgoals. This makes GSP more broadly applicable, as shown on Delivery (93% vs. 12%) and The Witness, where relabeling cannot be applied at all.
>
> **On the experiments**
>
> We have added ablations, with more detail in Review 4, and will include a deeper discussion of domain-specific successes and failures in the final paper.
>
> **Summary**
>
> Full search-tree updates are what let GSP learn policies that are directly executable greedily in very sparse-reward domains across a wide variety of problems, something no prior method achieves uniformly. We believe that this, together with the specific novelties outlined above, constitutes a contribution beyond incremental engineering.

---

> > ### Author Rebuttal · Reviewer_jNZG · 2026-04-02
> >
> > I thank the authors for their informative response. LRTA* must have been my confusion then. Overall, I however tend to stay with my original assessment. The results of the paper are interesting, but the paper doesn't really contain enough novel contributions for a conference paper over previous works. As mentioned in another review, the curriculum style learning has already been studied by Zilles et al (AIJ11), learning Q values / heuristic estimates from search output has also been considered before (e.g., Ferber et al ICAPS22). Moreover, the general architecture is close (actually identical) to the works by Ståhlberg and Geffner.

---

> > > ### Author Response · Authors · 2026-04-07
> > >
> > > We thank the reviewer for the follow-up. We are pleased to hear that the concerns raised in the original review are now considered fully resolved. Yet, we are somewhat surprised that the current score remains unchanged, since the remaining criticism appears to concern mainly the interpretation of novelty rather than any unresolved technical issue.
> > >
> > > On novelty, we agree that the cited prior work is relevant. However, we believe the present contribution is more distinct than the reviewer suggests.
> > >
> > > First, Arfaee et al. (2011) rely on backward random walks from the goal to generate progressively harder training instances, and explicitly note that this requires predecessor access for uninvertible operators and applies only to single-goal settings. Ferber et al. (2022) in turn state explicitly that two of their three methods are inspired by Arfaee et al. and likewise generate training states by increasingly longer backward walks from the goal; for partially specified goals they use FDR regression. Their third method is an approximate value iteration variant inspired by Agostinelli et al. (2019).
> > >
> > > By contrast, our method does not rely on backward walks from known goal states, predecessor access, invertible operators, or regression-based state generation. The learning signal is obtained directly from standard forward best-first search. This difference is important because it removes the need for goal-rooted backward instance generation and makes the method applicable under the basic forward-search model alone, including conjunctive-goal settings and domains with non-invertible actions.
> > >
> > > Second, the contribution is not merely that we use WA* instead of real-time search. The key novelty is the training procedure built around the full forward search tree. In particular, GSP learns from the expanded frontier rather than only from successful traces, assigning signal also to unproductive branches. This yields a self-improving search-and-learning loop that directly improves action guidance for future instances.
> > >
> > > Finally, regarding the relational architecture, we do not present the architecture itself as a novelty claim; this connection to prior work by Ståhlberg and Geffner is stated explicitly in the paper. Our claim is that the contribution lies in the proposed learning/search framework: replacing real-time search and relabeling-based learning with a forward best-first search loop that learns directly from search data and generalizes across instances, goals, and problem sizes.

---

### Official Review · Reviewer_AHjM · 2026-03-11

**Soundness:** 3
**Presentation:** 2
**Significance:** 3
**Originality:** 3
**Overall Recommendation:** 5
**Confidence:** 4

**Summary:**

This paper proposes GSP to learns heuristics for classical planning by combining WA* search with Q-learning. A relational GNN parameterizes Q(s,a), which guides WA* search on training instances; the resulting solution paths and dead-ends are used to update Q via Q-learning. The key contribution is that this loop generalizes across planning instances of varying size: the same heuristic trained on small instances guides search on much larger instances. Experiments span IPC 2023 benchmarks, combinatorial puzzles (Sokoban, The Witness, 24-Puzzle), and PushWorld, showing competitive or superior performance to existing methods including Lifted HER, LAMA, and specialized puzzle solvers.

**Compliance With Llm Reviewing Policy:**

Affirmed.

**Final Justification:**

Rebuttal has addressed my concerns

**Key Questions For Authors:**

1. How does GSP perform with standard planning heuristics (e.g., h_ff, h_max) plugged into the same WA* framework, without any learning?
2. Have you measured whether the learned heuristic is approximately admissible?
3. What happens with uniform instance sampling instead of the unsolved/solved/satisficed strategy?
4. Could you please explain why MCTS with a learned value function would be infeasible in this setting?

**Limitations:**

yes

**Strengths And Weaknesses:**

Strengths:

- The core idea of replacing RL's real-time search with best-first search for learning generalizing heuristics is well-motivated and clearly presented.
- The size generalization results are impressive; these are genuinely hard for RL methods.
- The relational GNN architecture for Q-functions is well-designed. Encoding action choices as auxiliary action-atoms within the relational input is elegant.
- The breadth of evaluation is a strength--IPC benchmarks, puzzles, and PushWorld cover diverse planning challenges and the method uses identical hyperparameters throughout.
- The paper is upfront about failures. The 24-Puzzle result (0% coverage) and the analysis of why (fixed budget insufficient for initial signal, GNN overhead vs. fixed-size MLPs) is appreciated.

Weaknesses:

- The baseline comparison has significant gaps. Are there standard, domain independent heuristics that you can compare against? LAMA seems to use landmarks, but simpler heuristics would clarify what the learned heuristic contributes beyond well-known planning heuristics. - The novelty-based exploration methods cited in related work are never compared against. And no MCTS-based baseline is included despite extended discussion of MCTS limitations.
- The dismissal of MCTS in Section 2 is under-argued. The claims that MCTS is "ill-suited for single-goal pathfinding," "does not inherently support size generalization," and "struggles in non-adversarial puzzle domains" conflate limitations of random rollouts with limitations of MCTS as a framework. MCTS with a learned value function (as in AlphaZero) sidesteps the rollout issue, single-player MCTS is well-established, and size generalization depends on the value function architecture (the GNN), not the search algorithm. A more careful discussion or an empirical comparison would strengthen the paper.
- Several key design choices are not ablated. The instance selection strategy is presented as important but never compared against uniform sampling or alternative curricula. The search-derived lower bound mechanism (taking max of search return and bootstrap target) is claimed to stabilize learning but is not ablated.
= There is no analysis of what the learned Q-function captures. Is it approximately admissible? How does it relate to known heuristics? When does it over- or under-estimate? Understanding the learned heuristic's properties would provide insight beyond the empirical coverage numbers.
- The WA* search at test time sometimes hurts performance compared to greedy execution (Blocksworld: 100%→79%, Spanner: 100%→27%, Transport: 73%→57%). The paper attributes this to large branching factors but doesn't analyze this failure mode in depth. If the learned heuristic is good enough for greedy execution but degrades under systematic search, that suggests something about the heuristic quality worth investigating.

---

> ### Author Rebuttal · Authors · 2026-03-31
>
> We thank the reviewer for their detailed review and constructive feedback. We address the author's questions below.
> ## MCTS is less suitable in our setting
> We agree that the original discussion of MCTS was too brief. Our intent was not to claim that MCTS is fundamentally inapplicable to classical planning, but rather that, in the setting considered here, it appears less suitable than search guided by a learned heuristic.
>
> The distinction is as follows. A* search maintains an explicit frontier and expands nodes according to the cost-so-far and heuristic cost-to-go. This makes the search objective directly aligned with shortest-path planning. By contrast, MCTS is primarily a root-centered decision procedure: it improves action selection at a current state through repeated simulations and value backups. In large combinatorial planning domains with sparse rewards, often a single goal state, and high branching factors, discovering reward signals through simulations becomes unlikely.
>
> AlphaZero alleviates the random-rollout issue. However, prior work has already reported that policy-value MCTS can struggle in puzzle-like domains: Orseau & Lelis (2021) found success only on The Witness. Moreover, those experiments concern fixed-size generalization which is substantially easier than our size generalization.
>
> While we did not exhaust every possible improvement to AlphaZero, our results (same budgets as main experiments) suggest that it is less effective than GSP for our goals:
>
> | domain | greedy (%) | search (%) |
> |---|---:|---:|
> | blocksworld | 31 | 0 |
> | childsnack | 0 | 0 |
> | ferry | 19 | 0 |
> | miconic | 28 | 0 |
> | rovers | 0 | 0 |
> | satellite | 0 | 0 |
> | spanner | 73 | 71 |
> | transport | 0 | 0 |
>
> 'search' is also worse than 'greedy' indicating the value function hurts performance, unlike the policy (see reasons below). Size generalization requires learning how to act from almost any part of the state space, not only from the trajectories actually executed. MCTS concentrates computation along simulated paths from the root. In contrast, A\* constructs a broad search tree around promising states, including alternatives, dead ends, and lower-bound information of solution paths. This provides richer supervision for learning a heuristic that generalizes.
> ## Greedy vs. WA*
> We agree that the difference between greedy and WA* performance deserves a clearer explanation. Our explanation is that a) the learned heuristic is substantially stronger as a local ranking mechanism than as a global value function on out-of-distribution data, and b) huge branching factors in test instances.
>
> a) On training problems the learned heuristic eventually scores well enough globally that the number of expanded nodes is close to or equal to the solution length, indicating that the search is nearly perfectly guided. However, this property deteriorates more quickly than the relative ranking of actions within a state when generalizing to larger or structurally different test instances. In result, greedy search can remain effective because it only relies on choosing the best local action, whereas WA* must order a frontier using global poorly calibrated scores.
> This is due to placing the state together with all applicable actions into a joint relational graph, allowing the GNN to score the actions in direct context to one another which benefits action selection within a state. However, this design does not encourage well-calibrated scores across states.
>
> b) IPC test instances are categorized as easy, medium, or hard with drastic increases in branching factors from one category to the next. They are partly designed to directly challenge (A*) search algorithms.
>
> We will clarify these points in the revision.
> ## Ablations & Admissibility
> For ablations see the response to Reviewer 4. Admissibility is not enforced during training and solutions indicate that the heuristic is not admissible.
> ## Standard heuristics
> Results for the requested domain-independent heuristics on IPC data (same search budgets as main experiments):
>
> | Domain | WA*\_hff Cov. (%) | WA*\_hmax Cov. (%) |
> |---|---:|---:|
> | blocksworld | 12 | 3 |
> | childsnack | 0 | 0 |
> | ferry | 67 | 6 |
> | floortile | 8 | 0 |
> | miconic | 100 | 20 |
> | rovers | 31 | 6 |
> | satellite | 67 | 6 |
> | sokoban | 34 | 23 |
> | spanner | 33 | 32 |
> | transport | 34 | 7 |
>
> These results show a gap between learned domain-specific and domain-independent heuristics. Miconic is the exception, where h_ff attains full coverage. Ferry and satellite also remain fairly tractable for h_ff. In most other domains, however, the domain-independent heuristics solve only a limited fraction of the benchmark, typically the easiest instances.

---

> > ### Author Rebuttal · Reviewer_AHjM · 2026-04-03
> >
> > - MCTS: thanks for the discussion. I have a couple of questions: (1) did you use the same Q-function for MCTS as you did with your planner? (2) How did you select the root node for the MCTS search? Since you are in the planning setting, I imagine you have to commit to some search-control strategy, but I cant figure out how you did that for your MCTS experiment.
> > - Comparison against standard heuristics: this is a very nice comparison and is showing the value of the learned heuristics beyond general domain-independent ones; I would suggest adding these results to the paper.
> >
> > Reflecting on my review, I think my questions about MCTS are more of a curiosity because you have convinced me that the real-time search approach doesn't totally make sense for offline planning, where you want to expand nodes that will best improve your value estimates. So I am happy to increment my rating to a full accept.

---

> > > ### Author Response · Authors · 2026-04-04
> > >
> > > We thank the reviewer for the thoughtful follow-up and for the positive reassessment of our submission. We are glad that our clarifications could resolv the concerns.
> > >
> > > Apologies for the lack of clarity regarding the MCTS experiment; the character limit prevented us from including further details. Regarding the questions:
> > >
> > > 1. Yes, we used the same $Q$-function architecture as in the main paper based on relational state and goal descriptions.
> > >
> > > 2. In the greedy setting, we executed the learned policy $\pi_\theta$ greedily from the initial state $s_0$ onward on the test instances. The “search” setting should not be confused with our A* search. Here, “search” means that at each current state $s$, we performed an MCTS lookahead rooted at $s$ with a budget of 100 simulations in order to improve action selection. After these simulations, the next action was selected at the root based on the child visit counts, yielding the successor state $s'$. The process was then repeated from $s'$. When available, we reused the subtree rooted at $s'$ from the previous lookahead.
> > >
> > > Adding the comparison to domain-independent heuristics does help place the experimental results in context, and we will include these results in the paper.

---

### Official Review · Reviewer_jZGS · 2026-03-13

**Soundness:** 4
**Presentation:** 3
**Significance:** 4
**Originality:** 4
**Overall Recommendation:** 5
**Confidence:** 2

**Summary:**

The authors present a new method GSP to generalize across instances (states, goals, problem size) of challenging combinatorial problems with goal states. The crux of the method is to (1) learn a Q function to do search with WA*, a weighted cost-to-go version of A* (or best-first search), and then update the Q function as a result of the search and (2) use relational GNNs. The search expands on the most promising nodes and, upon reaching a goal, gives a lower bound for the bootstrap return. The authors present approximately SOTA results on the IPC learning benchmark with significantly less node expansions than other SOTA algorithms.

**Compliance With Llm Reviewing Policy:**

Affirmed.

**Final Justification:**

I maintain my recommendation of acceptance for the paper after reading the rebuttal, the other reviews, and the response to other reviews, though do not have high confidence in my rating due to the paper being out of my domain.

The usage of best-first search with learned Q values is interesting. Even if no independently new ideas are presented, I do think a thorough evaluation of a SOTA method that combines proposed/older methods with more modern methods (e.g. learned Q value guided best-first search) is valuable, and satisfies ``Does this work offer a novel combination of existing techniques, and is the reasoning behind this combination well-articulated?'' in the Originality category.

That being said, this is out of my domain, so I maintain a low confidence of 2 in my rating.

**Key Questions For Authors:**

- Are the Q_{\theta}(s,a) for fixed goals in that particular problem instance? Then, does GSP naturally zero-shot generalize what counts as a goal-state given a new state (without ever explicitly giving the GSP an example of that goal state in the past) in the Blocksworld task?

- Are there a critical number of solves required before GSP can do zero-shot transfer?

**Limitations:**

yes

**Strengths And Weaknesses:**

Overall, a well-written paper presenting a new method GSP with strong generalization results in tough combinatorial benchmarks.

Strengths:

- The method presentation of learning a Q function to search via best-first search, and searching updating the Q function is straightforward.

- The message-passing formulation of the GNN is carefully walked through, and design decisions are clear.

- The results of the method are quite strong, and exhibit impressive generalization over problem size with a small exploration budget. The results are SOTA.

Weaknesses:

- Would like a better understanding of conditions under which zero-shot generalization emerges and the importance of that result -- e.g., what is the minimum number of instances required at Level 0 before GSP is able to zero-shot generalize?

Minor:

- Clarifying the problem statement -- does GSP learn one Q_{\theta} over *all* examples? Or is the Q_{\theta} learned from scratch for every new problem instance?

- ``Importantly, MLPQ is shared across all action schemas...rather than through schema-specific output heads'' -- may be good to explicitly write why this is desirable for GSP.

---

> ### Author Rebuttal · Authors · 2026-03-31
>
> # Response to Reviewer
>
> We thank the reviewer for the careful reading and for the positive assessment of the paper. Below, we address each point of the comments and questions.
>
> ## Conditions under which zero-shot generalization emerges
>
> In our setting, zero-shot transfer onto test problems is the primary goal of training and results naturally from learning a goal-conditioned action-value function that exploits relational structure shared across problem instances. The model receives both the current state and the goal specification for each instance, at training and at test time, and therefore learns how actions contribute to goal achievement across varying object sets, initial states, and goal compositions.
>
> Regarding the reviewer's question about the minimum number of Level-0 instances required:
>
> Unfortunately, we do not have a clear number of instances required for generalization to Level 1 to succeed. It is noteworthy that the number of Level-0 training instances is already limited and, thus, the observed generalization is not a result of massive amounts of data.
>
> ## Clarification of whether one $Q_\theta$ is learned across all examples
>
> Yes. GSP learns a single shared parameterized Q-function $Q_\theta$ across all training instances within a domain, not a separate Q-function per problem instance. At test time, the learned $Q_\theta$ is applied directly to unseen instances without retraining.
>
> ## Whether $Q_\theta(s,a)$ is defined for a fixed goal in the current instance
>
> Yes. $Q_\theta(s,a)$ is goal-conditioned: the input includes the current state together with the goal of the current planning instance. Perhaps better denoted as $\tilde s = \{s, g\}$ where s is the state and g is the goal description, and $\tilde s$ being the goal-extended state description. Therefore, for a fixed instance, the score $Q_\theta(s,a)$ is evaluated relative to that instance's goal. This also answers the reviewer's Blocksworld question: GSP is not given past examples of the exact same goal state. Instead, it must generalize to new goal configurations by processing the symbolic goal description of the new instance and relating it to the current state.
>
> ## Is a critical number of solves required before zero-shot transfer appears?
>
> We agree that this is an important question. Our current results show that strong zero-shot transfer can emerge after training on relatively small instances. Transfer capabilities emerge from every solved instance, which enables solving increasingly larger and more complex instances. This bootstrapping is completed once all training problems are solved. More generally, our empirical observations suggest that the larger the training set, the more potent the generalization becomes, up to model-capacity limits.
>
> ## Why sharing MLPQ across action schemas is desirable
>
> We will make this motivation more explicit. Sharing MLPQ across action schemas encourages schema-agnostic scoring based on relational context rather than schema-specific output heads. This is desirable for GSP because it improves parameter sharing and reduces the risk of overfitting to schema identities, thereby supporting transfer across instances with different object combinations and grounded actions. In other words, the model learns a common scoring principle for grounded actions instead of learning separate schema-specialized predictors.
>
> We hope this clarifies the reviewer's questions.

---

> > ### Author Rebuttal · Reviewer_jZGS · 2026-04-01
> >
> > Thank you for your response. I maintain my recommendation of acceptance for the paper after reading the rebuttal, the other reviews, and the response to other reviews, though do not have high confidence in my rating due to the paper being out of my domain.
> >
> > I suggest that the authors make this goal-conditioned/joint state-goal parameterization explicit by emphasizing that the goal is included within the state. This is particularly relevant to connect some of the work to more general goal-conditioned RL problems in e.g. the continuous setting, that may benefit from some of the ideas and methods presented in the paper.

---

### Decision · Program_Chairs · 2026-04-30

**Decision:**

Accept (regular)

**Comment:**

Some reviewers express weak confidence and I agree with the authors' rebuttal regarding many responses.  However, I do particularly share concerns with one reviewer that the proposed ideas are simply learning Q-functions in the manner of relational GNNs, the latter of which have been used quite heavily in recent work on generalized planning.  That said, I am not aware of works have taken a Q-learning approach that allows GNNs to improve online from the "self-improving learning cycle" and it does seem important to highlight the value of relational GNNs in this setting.  The results reported in the paper are quite strong.  Overall, I believe this work represents a fundamental technique and performant contribution that make it a strong candidate for ICML.